# Family Facilitators of, Barriers to and Strategies for Healthy Eating among Chinese Adolescents: Qualitative Interviews with Parent–Adolescent Dyads

**DOI:** 10.3390/nu15030651

**Published:** 2023-01-27

**Authors:** Kiki S. N. Liu, Julie Y. Chen, Kai-Sing Sun, Joyce P. Y. Tsang, Patrick Ip, Cindy L. K. Lam

**Affiliations:** 1Department of Family Medicine and Primary Care, The University of Hong Kong, Hong Kong, China; 2Department of Family Medicine, The University of Hong Kong Shenzhen Hospital, Shenzhen 518053, China; 3JC School of Public Health and Primary Care, The Chinese University of Hong Kong, Hong Kong, China; 4Department of Paediatrics & Adolescent Medicine, The University of Hong Kong, Hong Kong, China

**Keywords:** healthy eating, adolescents, family factor, facilitators, barriers, KAP, qualitative

## Abstract

Healthy eating is vital in preventing obesity and long-term non-communicable diseases. This study explores potential family facilitators of, barriers to and strategies for healthy eating among adolescents in Chinese families to guide the development of effective interventions in the future. Parent–adolescent dyads were purposively sampled by age, gender, fruit and vegetable intake and household income. Key family factors were identified by thematic analysis. Fourteen themes under five domains were identified: family health with (1) illness experienced in the family; parental knowledge of (2) dietary recommendations, (3) the preparation of healthy food and (4) healthy food choice; parental attitudes towards (5) the importance of healthy eating and (6) the priority of family health; socioeconomic factors of (7) time concerns and (8) cost concerns; and food parenting practices in (9) nutritional education, (10) role modeling, (11) food provision, (12) child involvement, (13) parental supervision and (14) the cultivation of food preference. Useful strategies included incorporating healthy ingredients in adolescents’ favorite recipes and providing a variety of fruit and vegetables at home. There is a need to empower parents with practical nutrition knowledge, to be more authoritative in food parenting practices, to discuss healthy eating with children and to acquire practical skills related to time- and cost-saving healthy cooking.

## 1. Introduction

Healthy eating is vital in reducing adolescent obesity and long-term non-communicable diseases [1]. Unfortunately, the adherence rates to the dietary recommendations of fruit and vegetable (FV) and salt intake have been less than 15% in this age group [2,3], which calls for investigation into factors influencing their eating habits. 

Family is amongst the major agents that influence the home food environment of adolescents. Family factors incorporate two constructs. The first construct is parent characteristics, such as illness experienced, knowledge, attitudes and socioeconomic factors [4]. These personal attributes explain their intention to act and drive their practices. Health conditions in the family can promote discussion of diet-related health risks and the perceived importance of healthy eating [5,6]. A recent study also showed an increased level of nutritional knowledge in Greek children aged 10 to 12 years whose parents had diabetes and dyslipidemia [7]. Parental knowledge and attitudes have a positive impact on the eating habits of adolescents [8,9,10], although the underlying pathway is still unclear. Socioeconomic status enhances diet quality in the general population [11], including adolescents [12].

The second construct is the diet-related behaviors of parents that guide the eating habits of adolescents, often referred to as food parenting practices in the literature. Nutrition education, role modeling and food accessibility are found to be the most influential of these on adolescent food consumption [13,14,15]. Apart from these practices, there are other parenting behaviors that are adopted by the parents. A review by Vaughn et al. summarized nineteen parenting practices [4]. Among them, meal and snack routines, food availability, food accessibility and food preparation are practices related to food provision. Monitoring, encouragement and negotiation often exist as a chain of actions to prompt and guide healthy eating in adolescents. These, along with rules and limits with reasoning, act as parental supervision that serves as guidance to the adolescents on the expected eating behavior.

Parenting styles lead to a varied range of strictness and warmth in food parenting practices [16]. For instance, parents with an authoritarian style use coercive control to exert pressure on adolescents’ behavior without regard for their feelings. Authoritative parents guide adolescents’ practices by both rules and supervision while also responding to their needs. Permissive and uninvolved parents have unstructured practices characterized by indulgence and neglect with limited control of adolescents’ eating habits. An authoritative parenting style has been shown to promote healthy eating and reduce child obesity [17,18].

In a recent review of qualitative studies, we identified the influence of parent characteristics and food parenting practices on adolescent knowledge, attitudes and practices (KAP) of healthy eating [19]. The application of the KAP framework was useful in exploring understudied aspects, such as how food preference was formed in a family context. The review also showed there was a lack in the literature regarding Asian countries, especially concerning works involving multiple informants [19]. This study aims to explore family facilitators of, barriers to and strategies for healthy eating among adolescents in Chinese families. Drawing on the perspectives of both adolescents and parents provides better-informed guidance for future interventions in the local setting.

## 2. Materials and Methods

### 2.1. Subjects

Family dyads were sampled from families who had enrolled in a comparative cohort study to evaluate an empowerment program, the Trekkers Family Enhancement Scheme (TFES), targeting low-income families [20]. Health and social empowerment programs, such as classes of physical activities and nutrition workshops, were provided for voluntary participation by TFES families. Eligible families were those with adolescents aged 10–19 years, both parents and adolescents able to speak Cantonese, and parents being the primary caregiver of the adolescent.

The dyads were selected using stratified purposive sampling based on the daily servings of FV consumed, age and gender of the adolescents, as well as family household income and attendance at previous nutrition workshops. This aided in collecting a wide range of perspectives on the study topic. Adolescents were regarded as “healthy” if they had at least 5 servings of FV each day, “average” if they had 3 or 4 servings and “unhealthy” if they had 2 servings or less. Selected families were invited to join the study by phone and provided written consent before the interview. We first contacted 135 eligible families by phone, among whom 30 families did not answer the phone, 57 families refused to participate due to the lack of time or interest and 26 families did not get back after the first contact. We recruited three additional families on-site at a TFES event. In total, 25 families participated in the study. Each family received supermarket coupons for HKD 100 to compensate for their time after the interview.

### 2.2. Data Collection

The interviews were conducted individually with each parent–adolescent dyad on Zoom, and all families except one turned on the camera. The facilitator (K.S.N.L.) explained the research aim to the families at the beginning of the interview and conducted the interview using a semi-structured interview guide (Appendix A). The interview guide was developed from the KAP framework that has been used in previous studies [19,21]. Questions were first directed to the adolescents to elicit their perspectives before supplementation by the parents. The interviews lasted for 30 to 60 min and were conducted in Cantonese. We completed the 25 interviews between May 2020 and March 2022, and data saturation was reached when no new codes were identified after 3 consecutive interviews.

### 2.3. Data Analysis

Thematic analysis was conducted to identify themes of family factors [22]. Following the existing guide [22,23], we familiarized ourselves with the data, generated initial codes and formulated, reviewed and defined the themes. The interview recordings were transcribed verbatim in Chinese and analyzed using the NVivo software. Two trained research assistants did the coding independently, and any inconsistencies were resolved among the coders and the research team. Two authors (J.Y.C. and C.L.K.L.) reviewed the coding tree iteratively. The findings on family factors were grouped under parent characteristics and food parenting practices. Parent characteristics were further categorized into the domains of family health, parental knowledge and attitudes and socioeconomic factors.

The ethics of this study were approved by the institutional review board of the University of Hong Kong/ Hospital Authority Hong Kong West Cluster (reference no: UW 20-007).

## 3. Results

### 3.1. Subject Characteristics

Twenty-five adolescents (aged 12–19 years) and their mothers (aged 40–56 years) joined the interviews; one father joined the conversation midway through the interview. Among the adolescents, thirteen were female and eight consumed at least five servings of FV a day. Table 1 shows the distribution of subject characteristics, and Table 2 lists the characteristics of each family.

### 3.2. Family Factors of Adolescent KAP of Healthy Eating

A total of fourteen themes under five domains were identified from the interviews. The themes are illustrated by quotes from the interviews. The source of each quote is indicated by parent (P) or adolescent (A), family number, gender and age. More quotes were captured in the Appendix A.

#### 3.2.1. Influence of Parent Characteristics

Table 3 shows the eight themes grouped under family health, parental knowledge and attitudes and socioeconomic factors and how they influenced the KAP of healthy eating in adolescents.

Family Health—Illness experienced in the family (theme 1)

Some parents experienced positive outcomes from healthy eating, which were witnessed by and positively impacted their adolescents. This influence was more significant in families whose parents had health problems. In one family, the mother was obese with a family history of heart disease, which led the daughter to perceive her own susceptibility and adopt healthy eating in a way to promote eating habits in her mother.

“[Daughter] is scared when seeing me fat. Because my father died from a poor heart, and my heart is not good as well… She always asks me to control… my eating, asks me to [eat] lighter. Sometimes I cannot change, so she is very concerned about healthy eating.”(P21, F, 55)

2.Parental Knowledge—Dietary recommendations (theme 2)

Some parents had knowledge of the recommended daily vegetable intake and cited close to half catty, i.e., 300 g, per person; one mentioned the 3-2-1 proportion of grains, vegetables and meat in a lunchbox. They tried to prepare meals according to the dietary recommendations they knew.

“Yes [will follow], for a lunchbox, two grids for vegetables, one for meat, and three for rice, isn’t it? [Daughter] has been taught previously.”(P9, F, 40)

Conversely, parental knowledge deficiency about dietary recommendations was a barrier in many families. The parents generally underestimated the recommended servings of FV and thought having one fruit and one bowl of vegetables daily was sufficient, which then justified the insufficient consumption in their adolescents. Some parents did not know that non-leafy vegetables such as carrots and gourds were vegetables and perceived difficulty in eating three servings of vegetables a day. This knowledge and perceptions tended to be passed down to their adolescents in most families.

“Corn, potato, sweet potato, these are not vegetables, [they] are [a] sort of starch.” (A11, F, 13) “It is already enough when sometimes cook pumpkin, then stir-fry [a dish of] leafy vegetables and another [dish of] meat. Then [there are] insufficient leafy vegetables if [we] need to eat two to three servings.”(P11, F, 44)

3.Parental knowledge—Preparation of healthy food (theme 3)

Despite some deficiency in knowledge of dietary recommendations, most families regarded home-prepared meals as healthy due to the limited use of salt, seasonings and oil. They generally knew that boiling and steaming were healthy cooking methods. Some parents also used other low-oil methods, including stir-frying and braising, to prepare dishes with different types of vegetables. This tended to improve their adolescents’ knowledge and preference towards a variety of vegetables.

“[Mum] cooks with chicken, vegetables, red bell pepper, and those mushrooms… Mainly boiling and stir-frying… sometimes braising… [I] sometimes buy those tomatoes, eggplants and… potatoes [for takeaway].”(A10, F, 14)

Several parents described how they balanced health and the taste preference of their adolescents to provide healthy meals. For example, they modified recipes their adolescents liked by adding nutritious ingredients such as fish and vegetables.

“[We] have tried the vegetable cake at IKEA and found it tasty, [children] also love it. There should be some broccoli, potatoes… [those carrots and corns.] Her father added some cheese… pumpkin…when he made himself.”(P19, F, 55)

A parental lack of knowledge on how to prepare tasty meals using less oil and seasonings was a barrier. Their adolescents regarded home meals as bland. They might not eat or eat little during the main meals and tend to eat unhealthy instant noodles or takeaway food instead.

“I do not use many of those [seasonings]. [Children] always say that [my dishes] are tasteless, that means bland. But I do not want to add too many seasonings… If the dishes do not match their taste, they will not eat much and are hungry after a few hours. Sometimes [I] ask [son] to cook noodles, sometimes he wants to buy [chicken] congee [downstairs].”(P13, F, 48)

Parents usually used sugar, salt, soy sauce and chicken powder as seasonings. The lack of knowledge on healthy alternatives, such as spring onion, garlic, onion and herbs, to salty seasonings was another barrier to eating healthy home-prepared meals.

Another strategy that many parents used was making homemade drinks for their families, including herbal tea and soup. This reduced the consumption of prepackaged beverages that are high in sugar. Making soup with vegetables was a strategy to increase the intake of healthy food among adolescents.

“Make soup with white gourd when [I am] free… sometimes with yam, carrot and corn… or with peanut and lotus root.”(P25, F, 47)

4.Parental knowledge—Healthy food choice (theme 4)

A common barrier to healthy eating was a deficiency in knowledge of healthy choices when eating out. Most families perceived eating-out food as always oily and heavily flavored, and they could only reduce the practice rather than selecting healthy choices on the menu.

“Eating out like stir-frying flat rice noodles with beef [is] very fatty… It is not easy to choose healthy [dishes] when eating out. [We] will be more indulgent.” (P4, F, 56)

Another barrier was that parents often purchased instant, frozen and canned food to enable adolescent self-cooking. Most of these items were unhealthy, including instant noodles and luncheon meat.

“I will cook instant noodle if I want to eat… I will cook those meat balls or luncheon meat… [They] are available [at home], and dumplings, all are ready-to-cook and put into [the pot directly].”(A3, F, 14)

5.Parental attitudes—Importance of healthy eating (theme 5)

Many parents were concerned about their own health as they aged and how to reduce the risk of chronic diseases by healthy eating. They tended to adopt good food parenting practices to facilitate healthy eating in the family. Some parents also perceived the importance of healthy eating for adolescents, with a few specifically identifying that it can help prevent health problems such as being overweight, constipation, skin problems and maintaining energy balance, a traditional Chinese medicine concept.

“[Children] say they do not have constipation, not a problem, as they eat vegetables and fruit every day. [I] urge them to eat.” (P15, F, 46)

6.Parental attitudes—Priority of family health (theme 6)

Parental prioritization of health when making food choices was a facilitator of home cooking practices. Most parents preferred home-prepared meals over eating out due to the lower use of oil and seasonings, better freshness and hygiene. They also developed strategies to reduce unhealthy eating practices and to model healthy food choices for their adolescents.

“This son loves drinking coke very much, needs to drink one can every day. So, I try to buy other [beverages]… I buy chrysanthemum tea or milk, which are healthier. It does not matter even he drinks daily. The harm is less [compared to coke].”(P21, F, 47)

Health was not prioritized on some occasions, such as when eating out or snacking. Parents considered family taste preferences and chose eating-out dishes that were seldom served at home even though they were unhealthy.

“Will pick those [dishes] less likely to be served at home [when eating out]… like spiced salt [seasoned], deep-fried… [Children] like fried [food].”(P4, F, 56)

Some parents purchased the unhealthy snacks their adolescents demanded and used their adolescent’s infrequent consumption or normal body shape as justification.

7.Socioeconomic factors—Time concern (theme 7)

Time was a commonly mentioned barrier to home-prepared meals that resulted in eating out or buying takeaway meals. Some parents did not have the time or energy to prepare meals after taking care of the family or working long hours.

“I am not free to take care of [children], I am very busy for the whole day, so I do not cook as I have no energy [for this].”(P14, F, 49)

Some families highlighted that they ate out or opted for takeaway food at times because it was convenient to go to chain restaurants or fast-food shops for a quick meal.

“If not out of time, not really too busy, we will not eat out. Eating out in our definition is quick meal.”(P16, F, 48)

A lack of time in adolescents was also a barrier to healthy eating. One parent explained how she had to prioritize her adolescent’s school schedule when it conflicted with mealtime. She revealed that her daughter’s schedule was packed with schoolwork, and she only had time for a quick lunch with no time to eat fruit on school days.

8.Socioeconomic factors—Cost concern (theme 8)

Cost was a facilitator to limit eating out and snacking in most families. They regarded eating out as expensive and snacking as an unaffordable additional expense.

“Not [eating out], because sometimes… quite expensive to eat out with that many people in the family.”(P15, F, 46)

On the other hand, the higher cost of fresh meat led some parents to opt for cheaper frozen or ready-to-cook meat. Some supplemented that more seasonings were required to marinate frozen meat to remove the “refrigerator” taste.

“Best to eat [fresh meat] every meal, but you know how expensive it is… At least you need to remove the “refrigerator” taste of frozen [meat], need to add more… seasonings, that definitely is less healthy.”(P16, F, 48)

Food waste was another cost concern resulting in insufficient fruit intake in some adolescents. Their parents did not keep a stock of fresh fruit, especially a large volume, at home because they feared it rotting and going to waste before it could be consumed.

“I usually buy [fruit] when [family] requests, otherwise will be left till rotten… Afraid of food waste.”(P6, F, 42)

#### 3.2.2. Influence of Food Parenting Practices

Six themes under food parenting practices were identified, which are summarized in Table 4. A variety of parenting styles was exhibited, particularly in themes 12 to 14.

9.Nutrition education (theme 9)

Some parents improved their adolescents’ knowledge of healthy eating by educating them on the health outcomes of eating different foods. They often explained the consequences of eating junk food to guide their adolescents’ practices.

A few parents also provided suggestions on how to choose healthy options when eating out. They commonly advised no deep-frying, less gravy or sauce and avoidance of sugary drinks.

“Can only tell them, ‘When making food choice, do not pick those pan- and deep-fried food, choose something lighter, or not that greasy.’”(P19, F, 55)

Not every family discussed food and nutrition-related issues. Some parents did not bring up this topic because health might not be the first priority when making decisions on food. Some did not perceive the need thinking their families already had healthy eating habits.

10.Role modeling (theme 10)

Most adolescents adopted the eating habits of their parents, which were considered to be the family norm by the dyads. Parents tended to model healthy eating habits such as limited intake of oily food and snacks. When this occurred, acceptance and preference for these habits also increased in adolescents.

“[I] do not drink [packaged drinks] since a child… Believe it is unhealthy to drink… Yes [influenced by mum], mum almost not eating these [snacks].”(A10, F, 14)

The parental practice of not eating fruit daily was a barrier to the development of the adolescents’ positive attitudes and practice of adopting such a healthy eating habit. Fruit was usually consumed together with the family and prepared by parents. In families whose parents did not have the habit of daily fruit intake, most adolescents would also have a low fruit consumption because they were not reminded nor provided with ready-to-eat fruit.

“Very rare [for me] to eat fruit… Not specifically like eating fruit… Not cutting everyday… maybe once [a week].”(P24, F, 40)

11.Food provision (theme 11)

Most families illustrated how parents influenced the eating habits of adolescents by controlling food provision at home. One common example was having regular home-prepared meals, which cultivated the adolescents’ taste preference and countered the unhealthy habit of eating out.

“Mum used to deliver lunchbox [to us] since a child… I continue cooking at residential hostel in the university, probably due to [the habit of] eating home-prepared meals since [I was] small.”(A1, F, 18)

Another example was the variety of FV at home. Some families specified having a variety of vegetables for home meals or stocking different types of fruit for serving at home. These practices increased the interest in consuming FV in adolescents.

“Buying different types [of fruit] for [daughter] to eat… For example, not keep eating apples daily for the whole week, like eating apple today, dragon fruit tomorrow, and banana the day after… Because once eating apples on consecutive days… and she complained.”(P9, F, 40)

The availability of ready-to-eat fruit was frequently described as a facilitator of fruit consumption among adolescents. Most of them did not serve themselves and relied on their parents to peel and cut fruit into pieces for easy serving.

“[Son] eats as much as I prepare. If there is no fruit at home or not cut, he has nothing to eat. He will not actively take [the fruit] and eat on his own.”(P17, F, 52)

Parents also controlled the availability of snacks to reduce snacking habits in adolescents. Both parents and adolescents agreed that the presence of snacks at home would lower their self-efficacy in controlling snack consumption. It served as a temptation and increased their urge to eat them.

12.Child involvement (theme 12)

A few parents illustrated how they chose healthy food together with their adolescents when grocery shopping. It served as a chance to demonstrate knowledge of making food choices based on nutrition labels or claims on the package. They also shared with their adolescents the practical skills in buying takeaway, such as requesting low sugar when ordering drinks.

“Sometimes we read [nutrition label] together when [daughter] accompanies me to the supermarket. For example, we like certain brand of biscuit, and will compare the sugar, fat or whatever [on the label] among the flavors to make decision.”(P1, F, 51)

Involving adolescents in meal preparation was perceived as a strategy to enable adolescents to eat home-prepared meals when parents were not available to cook. They could gain cooking skills by observing how their parents cooked and participating in washing and cutting ingredients.

“[Son] has been preparing home meals for years… We are busy… basically back home late… He does the washing, in turn with his brother… Because sometimes when [we are] busy, they will cook on their own and serve first.”(P16, F, 48)

To facilitate healthy eating of home-prepared meals by adolescents, some parents ensured the availability of fresh ingredients and ready-to-cook food at home, such as vegetables and marinated meat, for easy and quick cooking.

“Sometimes when [children] are at home, [I] will marinate some pork, or chicken wings in advance. They will bake for a while in the oven for lunch. Side dishes are here anytime.”(P16, F, 48)

However, some families encountered barriers in engaging adolescents in meal preparation who ended up lacking cooking skills. They might need to prepare the meal quickly and did not have spare time to supervise their adolescents when cooking, or adolescents were excluded from the responsibility of food preparation.

“Usually, we cook very quickly when coming back from work, then will not bother to ask her to help with cooking. Because time is very tight, rushing to finish cooking and dine.”(P23, F, 48)

13.Parental supervision (theme 13)

Some parents set food rules to guide adolescents on healthy food consumption. The rules could be on limiting the servings of snacks or eating a certain amount of vegetables in a meal. Parents might supplement the rules with an explanation of the rationale, such as the need for a certain daily intake of vegetables.

“[I] tell [son] that ‘like you eat out in the afternoon, basically there are no vegetables, so you have to eat more at dinner at home.’”(P15, F, 46)

It was common for parents to monitor their adolescents’ eating habits and remind them about excessive snacking or insufficient vegetable intake. The act of prompting might also reinforce knowledge of health outcomes and the relevance of healthy eating in adolescents.

“[Daughter] will eat two or three pieces of chocolate. [I would] ask her not to eat that much, [it is] fatty… Need to remind her several times before she puts it down.”(P11, F, 44)

On the contrary, some parents did not supervise their adolescents’ eating habits, especially snacking. They adopted a permissive approach to let their adolescents decide what and how much to eat because they believed their adolescents were old enough to be sensible. These parents and adolescents often perceived the snack intake as not excessive though it might not be true.

“Nine individual packs [in a large bag of pretzel sticks]… usually finished all in two to three days… No, [mum will not supervise on snacking] because [I am] not eating frequently.”(A5, M, 15)

14.Cultivation of food preference (theme 14)

An important facilitator was to enhance adolescents’ preference for healthy food. Two parents explained their strategies for highlighting the positive attributes, such as the sweetness of vegetables and making it fun to eat FV.

“May instill into [daughter] since [she was a] kid, asking her to hold and bite a piece of vegetables, and telling her [it is] sweet, ‘Are these vegetables very sweet?’ Praising her when small, and she must follow you and said ‘Yes, yes, very sweet.’ Sometimes competing with her to see who finish eating vegetables faster.”(P11, F, 44)

Another facilitator was the consideration of the food preferences of the adolescents. Parents prepared meals in such a way as to encourage positive attitudes towards eating home-prepared meals and FV.

“[Son] has been fine with eating vegetables… He likes eating those vegetables… You will not buy those he dislikes, such as those with strong taste… I resisted those tastes when I was small. I am afraid that the kids will resist too… So I buy those common ones like Choy Sum, lettuce, broccoli, cauliflower.”(P22, F, 52)

## 4. Discussion

### 4.1. Major Findings

This study identified fourteen themes of family factors that could influence the adolescent KAP of healthy eating in working families in Hong Kong. The key facilitators included positive parental attitudes towards healthy eating, healthy food provision and parental supervision. The main barriers were deficiency in dietary knowledge in parents, time and cost concerns and limited family discussion on food-related issues. Role modeling had both positive and negative influences depending on parental eating habits. The families reported useful strategies using an authoritative style, such as incorporating healthy ingredients in adolescents’ favorite recipes, provisioning a variety of FV at home and involving adolescents in meal preparation.

#### 4.1.1. Parental Attitudes towards Healthy Eating

Parents, in general, have positive attitudes toward the perceived importance of healthy eating and the priority of health in food affairs. Studies have shown its facilitating effect on the diet quality of children [9] and adolescents [10]. Some parents related this perception to the public promotion in the mass media of healthy eating in NCD risk reduction, which can be further utilized to disseminate nutrition knowledge targeting parents.

#### 4.1.2. Food Parenting Practices

Parents focus on parenting practices related to food provision and supervision. They emphasized how they served regular home meals using healthy cooking methods, controlled the availability of unhealthy snacks and monitored and prompted food consumption. Nutritional education is uncommon as they seldom discuss food-related issues in the family. A recent study showed that family conversations about food during their childhood enhanced the eating habits in adults [24]. There is a potential benefit to adolescent attitudes that comes with raised awareness through regular discussion on the topic.

#### 4.1.3. Parenting Styles

Parenting styles have a distinctive role in modulating food discussion. For example, child involvement in food affairs is an authoritative approach to invite adolescents to contribute to the home food environment. It can increase the self-efficacy of children to eat healthily on their own [25]. Unfortunately, many adolescents have no responsibility for food preparation in Hong Kong. This is substantially lagged behind other western countries where one-third or more of children helped with meal preparation [25]. This might be explained by the cultural perception that mothers are responsible for housework, including cooking [26], and by the burden of academic work on adolescents [27].

Many parents using a permissive parenting style explained that the authoritarian approach—forcing their children to eat or not to eat certain foods—was not applicable to adolescents who have the maturity to decide on their own. This is comparable to the findings of a previous review that coercive control was more effective on younger children [15]. Conversely, adolescents with parents who cultivated their food preference towards home-prepared meals and FV from a young age can maintain healthy eating habits. This is in accordance with a longitudinal study in Australia that showed the diet quality of children with authoritative parents remained the best at both the ages of 4 and 14 years [28]. This further supports the need to adopt an authoritative parenting style in early childhood to supervise healthy eating and promote dietary KAP in adolescents.

#### 4.1.4. Parental Knowledge

There are gaps in parental knowledge on dietary recommendations, tasty meals with low salt and seasoning and healthy eating-out choices. The lack of knowledge on dietary recommendations was also noticed among adults in previous studies [29,30,31], which might indicate the global issue of insufficient public education on nutrition topics. Although there is evidence of improved dietary intake through family meals, it is unclear if the composition and quality of home-prepared meals matter. It is hypothesized that adolescents develop a negative preference towards healthy eating and tend to eat out when home-prepared meals are too bland.

Many parents perceive themselves as not knowledgeable enough and unable to provide a healthy eating environment to adolescents. They are uncertain if their meals provide sufficient nutrients to their children and what actual health benefits are brought. This could be linked to the limited nutrition knowledge covered in their education [32]. It is interesting that a few parents obtained the knowledge from the textbooks of their children, which suggests possible knowledge exchange in the family.

#### 4.1.5. Socioeconomic Factors

Time and cost concerns are the main family barriers to healthy eating practices in adolescents. Time concerns are more prevalent in disadvantaged families with one or both parents working and no one present at home to supervise the adolescents. The impact of time scarcity and low income on unhealthy eating habits has been demonstrated in previous studies [11,33,34,35]. It should be pointed out that these parents have a higher tendency to prioritize cost and convenience over health in food decisions. These attitudes would highly be passed down to adolescents [36], forming a vicious cycle.

### 4.2. Implications of Findings

This study highlights the necessity of addressing family factors to enhance food parenting practices and promote the adolescent KAP of healthy eating. Four strategies can be identified to facilitate and remove family barriers to healthy adolescent eating. First, empower parents with practical nutrition education on specific dietary recommendations, healthy recipes and healthy eating-out choices. Dietary recommendations should include portions and examples of food items, especially a variety of FV. Healthy recipes should be designed with simple steps, easily accessible ingredients and a variety of quick healthy cooking methods. Incorporating healthy ingredients in the usual recipes can be one solution. Healthy eating-out choices do not only refer to menu choices (avoid deep-frying) but also tips on ordering (request a separate serving of gravy) and serving (not drinking noodle broth).

Second, encourage parents to adopt an authoritative style in food parenting practices with particular emphasis on child involvement and cultivation of food preference. Parents should engage adolescents in grocery shopping and meal preparation, which helps them develop an interest in diet-related matters and promote healthy self-cooking. Positive descriptions of healthy food, such as sweet tasting and colorful, and considering adolescents’ preference in food choice are strategies to help them develop a preference for healthy food.

Third, promote more food discussions between parents and adolescents at home. Adolescents may be engaged to acquire knowledge on the recommended food servings and diet-related health outcomes, while parents can teach them how to prepare and cook the dishes. The dyads can collaborate in designing recipes that meet the dietary recommendations and also satisfy their taste preference.

Fourth, educate working parents on how to prepare healthy meals in a short time and on a limited budget. One strategy is to prepare ready-to-cook dishes in the morning to shorten the cooking time after work. This may also enhance the food flavor from the longer marinating time. Another strategy is to prepare dishes in larger portions for serving in two meals on alternative days. Bulk purchase of food usually earns a discount that helps save the budget. Boiling ‘quick’ soup with vegetables with or without meat may help as parents can attend to other tasks while cooking, and the ingredients can serve as a side dish. This also satisfies thirst and reduces the intake of sugary drinks during meals in adolescents.

### 4.3. Strengths and Limitations

This study has a strength in the use of a conceptual framework developed from the literature review that guided the analysis. This facilitated the exploration of family influence specific to each KAP construct. There was an in-depth discussion on the less-studied factors, such as family influence on adolescent attitudes, particularly food preference and self-efficacy.

Another strength was the interview setting of parent–adolescent dyads with enhanced validity. We anticipated that the adolescents might be less willing to talk or be honest in the presence of their parents and strategically invited them to speak before their parents and to start with easy questions on eating practices. During the interviews, we were pleasantly surprised that the adolescents openly described their socially “undesirable” attitudes and practices, such as their preference for unhealthy food and selecting fast food and unhealthy snacks on their own. The dyads supplemented with each other on food parenting practices. They shared some common thoughts on habitual practices in the family and prompted each other on food consumption. They also worked together in describing the influences of respective family factors. For example, some parents explained the low stock of snacks and how they monitored the eating habits of their children, whereas adolescents clarified how these family factors affected their KAP of healthy eating. The reliability of the conversation was assured by the consistency and interaction between the dyads that were observed in the interviews.

The results of this study need to be interpreted with the following caveats. First, the participants might have given socially desirable responses instead of what they truly thought or did. We hope this was minimized by the dyadic setting and non-sensitive topic, where over half of the family facilitators and barriers were agreed upon between the parents and adolescents. Second, nearly all the parents participating in the study were mothers, with a lack of opinions from fathers. We believe the information from mothers should reflect the general family situation since they are usually the parents responsible for home affairs and food preparation, especially in Chinese culture. Third, the subjects were participants of the TFES of a cohort study of working families whose characteristics and KAP of healthy eating may not be representative of families in Hong Kong. There may be potential volunteer bias. Future studies on families with wider socioeconomic backgrounds are required to substantiate the generalizability of our results. Fourth, the data collection period coincided with the COVID-19 pandemic, when the eating environment and practices might have been affected by social distancing; however, we believe the bias on the results should be minimal since the study explored long-term family factors.

## 5. Conclusions

This study identified fourteen themes of family factors that can support or impede adolescents’ KAP on healthy eating. Key facilitators are positive parental attitudes towards healthy eating, healthy food provision and parental supervision, while insufficient knowledge of dietary recommendations, healthy recipes and eating-out choices, time and cost concerns and limited food discussion in the family are the barriers. Useful family strategies to promote healthy adolescent eating are an authoritative approach: incorporating healthy ingredients in adolescents’ favorite recipes, keeping a provision of a variety of FV at home and involving adolescents in meal preparation. There is a need to resolve the barriers through public education to empower parents with practical nutrition knowledge, adopt an authoritative style in food parenting practices, promote family food discussion and acquire practical skills on time- and cost-saving healthy cooking.

## Figures and Tables

**Table 1 nutrients-15-00651-t001:** Distribution of subject characteristics.

Characteristics			Participants (%)	
Adolescents	Gender	Female	13	(52%)
		Male	12	(48%)
	Age (years)	Mean ± SD	14.84 ± 2.08	
		12–13	8	(32%)
		14–16	10	(40%)
		17–19	7	(28%)
	FV intake per day	Mean ± SD	3.6 ± 1.53	
		≥5 servings of FV (Healthy)	8	(32%)
		3–4 servings of FV (Average)	11	(44%)
		≤2 servings of FV (Unhealthy)	6	(24%)
	Gender	Female	25	(100%)
Parents	Age (years)	Mean ± SD	49.24 ± 4.65	
		40–49	14	(56%)
		50–59	11	(44%)
	Participation in nutrition workshop	Yes	10	(40%)
Household	Monthly income	Median	HKD 13,500–19,999	
		>HKD 27,000	4	(16%)
		HKD 20,000–26,999	5	(20%)
		HKD 13,500–19,999	8	(32%)
		<HKD 13,500	8	(32%)

FV = Fruit and vegetables, SD = standard deviation.

**Table 2 nutrients-15-00651-t002:** Characteristics of parent–adolescent dyads by family.

Family	Characteristics of		Household Income	Participation in Nutrition Workshop	Eating Status by FV Intake
	Adolescents	Parents			
1	F, 18 y	F, 51 y	0.75–1 median		Healthy
2	M, 16 y	F, 55 y	Below 0.5 median		Average
3	F, 14 y	F, 49 y	0.5–0.75 median	Yes	Average
4	M, 14 y	F, 56 y	Below 0.5 median	Yes	Average
5	M, 15 y	F, 55 y	0.75–1 median	Yes	Unhealthy
6	F, 14 y	F, 42 y	0.5–0.75 median		Unhealthy
7	F, 14 y	F, 51 y	0.75–1 median		Healthy
8	M, 14 y	F, 51 y	Below 0.5 median	Yes	Average
9	F, 17 y	F, 40 y	Above median	Yes	Healthy
10	F, 14 y	F, 48 y	0.5–0.75 median		Average
11	F, 13 y	F, 44 y	Below 0.5 median		Healthy
12	F, 18 y	F, 47 y	0.5–0.75 median		Unhealthy
13	M, 13 y	F, 48 y	Below 0.5 median	Yes	Unhealthy
14	F, 13 y	F, 49 y	Below 0.5 median	Yes	Unhealthy
15	M, 13 y	F, 46 y	0.75–1 median		Healthy
16	M, 12 y	F, 48 y	Below 0.5 median	Yes	Average
17	M, 15 y	F, 52 y	Below 0.5 median	Yes	Healthy
18	M, 19 y	F, 55 y	0.5–0.75 median		Average
19	F, 12 y	F, 55 y	0.75–1 median	Yes	Unhealthy
20	F, 17 y	F, 55 y	Above median		Healthy
21	M, 15 y	F, 47 y	Above median		Average
22	M, 13 y	F, 52 y	0.5–0.75 median		Healthy
23	F, 17 y	F, 48 y	Above median		Average
24	F, 13 y	F, 40 y	0.5–0.75 median		Average
25	M, 18 y	F, 47 y	0.5–0.75 median		Average

F = Female, M = Male, y = years of age, FV = Fruit and vegetables.

**Table 3 nutrients-15-00651-t003:** Influence of parent characteristics on adolescent KAP of healthy eating.

Family Factors				Influence on Adolescent		
Domains	Themes	Facilitators	Barriers	Knowledge	Attitudes	Practices
Family health	Illness experienced in the family	Witness positive health outcomes of healthy eating		+	+	
		Perceived risk of health problems			+	+
Parental knowledge	Dietary recommendations	Ensuring vegetable intake in daily meals	Uncertain about the recommended servings and definition of FV	-	-	+/-
	Preparation of healthy food	Healthy cooking methods and varied presentations of vegetables	Lack of knowledge regarding making tasty food that is low in oil and seasoning	+	+/-	+/-
		Balance of health and taste preference of adolescents in cooking	Lack of knowledge of healthy alternatives to salty seasonings		+	+/-
		Homemade drinks to replace prepackaged beverages				+
	Healthy food choice		Lack of knowledge of healthy choices when eating out			-
			Food available for adolescent self-cooking limited to unhealthy instant food			-
Parental attitudes	Importance of healthy eating	Belief in the impact of eating habits on own and adolescents’ health		+	+	+
	Priority of family health	Consideration of health in food choice	Consideration of taste preference over health when eating out or snacking		+	+/-
Socioeconomic factors	Time concern		Lack of time for home-cooking			-
			Convenience of eating out or takeaway food			-
			Consideration of adolescent school schedule over healthy eating habits			-
	Cost concern	Saving money by limited eating out and snacking	Choosing frozen or ready-to-cook meat for lower cost			+/-
			Concern about food waste prohibits keeping a stock of fresh fruit at home			-

“+” = Effect of facilitator, “-” = Effect of barrier.

**Table 4 nutrients-15-00651-t004:** Influence of food parenting practices on adolescent KAP of healthy eating.

Family Factors				Influence on Adolescent		
Domain	Themes	Facilitators	Barriers	Knowledge	Attitudes	Practices
Food parenting practices	Nutrition education	Education on health outcomes of eating habits	Limited discussion on food-related issues in family	+/-		
		Education on healthy eating-out choices		+		
	Role modeling	Parental practices of healthy eating habits	Parents not having a habit of eating fruit daily		+/-	+/-
	Food provision	Regular home-prepared meals			+	+
		Availability of a variety of FV at home	Unhealthy snacks available at home		+/-	+/-
		Serving ready-to-eat fruit				+
	Child involvement	Joint decision on healthy food choices during grocery shopping	Parents have little time to supervise adolescents in meal preparation	+/-		
		Involving adolescents in food preparation	Adolescents have no responsibility for food preparation	+/-		+
		Ready-to-cook food available for adolescents to cook				+
	Parental supervision	Setting food rules and explaining the reasons for the expectation	Lack of control or supervision of adolescents’ eating habits	+/-		+/-
		Monitoring and prompting food consumption		+	+	+
	Cultivation of food preference	Highlight positive attributes of FV, e.g., taste and fun			+	+
		Consideration of adolescents’ preference in preparing home meals and FV			+	+

FV = Fruit and vegetables, “+” = Effect of facilitator, “-” = Effect of barrier.

## Data Availability

All data generated or analyzed during this study are included in this published article and its Appendix A.

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
