# Peer review of "Family Facilitators of, Barriers to and Strategies for Healthy Eating among Chinese Adolescents: Qualitative Interviews with Parent–Adolescent Dyads"

_nutrients, 2023, doi:10.3390/nu15030651_

Round 1

Reviewer 1 Report

The paper titled: "Family facilitators, barriers and strategies of healthy eating among Chinese adolescents – Qualitative interviews with parent-adolescent dyads" is a very well conducted study; that use a qualitative research.   1. What is the main question addressed by the research? The question addressed by the research is the potential family facilitators, barriers and strategies of healthy eating among adolescents in Chinese families to guide development of effective interventions in future.   2. Do you consider the topic original or relevant in the field? Does it address a specific gap in the field? Looking for parents practices is very important to a well definition of interventions; as mentioned in the question addressed: family facilitators, barriers and strategies of healthy eating among adolescents in Chinese families to guide development of effective interventions in future.   3. What does it add to the subject area compared with other published material? Useful strategies included incorporating healthy ingredients in adolescents’ favorite recipes and providing a variety of fruit and vegetables at home. There is a need to empower parents with practical nutrition knowledge, to be more authoritative in food parenting practices, to discuss more on healthy eating with children and to acquire practical skills on time- and cost-saving healthy cooking.   4. What specific improvements should the authors consider regarding the methodology? What should further controls be considered?   Qualitative research. The methodology is clearly addressed.   5. Are the conclusions consistent with the evidence and arguments presented and do they address the main question posed? Yes, the conclusions are in accordance with the results and evidence presented.   6. Are the references appropriate? The references are appropriate, relevant, and in proper number.   7. Please include any additional comments on the tables and figures. A graphical abstract could be added.

Author Response

  1. The paper titled: "Family facilitators, barriers and strategies of healthy eating among Chinese adolescents – Qualitative interviews with parent-adolescent dyads" is a very well conducted study; that use a qualitative research. 1. What is the main question addressed by the research? The question addressed by the research is the potential family facilitators, barriers and strategies of healthy eating among adolescents in Chinese families to guide development of effective interventions in future. 2. Do you consider the topic original or relevant in the field? Does it address a specific gap in the field? Looking for parents practices is very important to a well definition of interventions; as mentioned in the question addressed: family facilitators, barriers and strategies of healthy eating among adolescents in Chinese families to guide development of effective interventions in future.   3. What does it add to the subject area compared with other published material? Useful strategies included incorporating healthy ingredients in adolescents’ favorite recipes and providing a variety of fruit and vegetables at home. There is a need to empower parents with practical nutrition knowledge, to be more authoritative in food parenting practices, to discuss more on healthy eating with children and to acquire practical skills on time- and cost-saving healthy cooking.   4. What specific improvements should the authors consider regarding the methodology? What should further controls be considered?   Qualitative research. The methodology is clearly addressed.   5. Are the conclusions consistent with the evidence and arguments presented and do they address the main question posed? Yes, the conclusions are in accordance with the results and evidence presented.   6. Are the references appropriate? The references are appropriate, relevant, and in proper number.   7. Please include any additional comments on the tables and figures. A graphical abstract could be added.

Response 1: Thank you for the positive feedback and suggestion. A graphical abstract of the fourteen family factors has been added to the submission platform.

Reviewer 2 Report

This manuscript examined family facilitators, barriers, and strategies of healthy eating among Chinese adolescents using qualitative methods. It is an interesting area of research; however, there are some parts that need clarification.

1.          How many potential participants were approached and how many accepted/ rejected?  How was the recruitment process? Please give more details on participants recruitment. Were they compensated?

2.         The authors mentioned that some participants were part of a larger study. What would be the implication in interpreting the findings besides the volunteer’s bias?

3.          The data collection period is during the Pandemic from 2020-2022. Is there any chance that this may affect the study's validity?

4.         From personal experience in interviewing adolescents, it was sometimes difficult to let the son/daughter talk openly and honestly in front of their parents. The authors said that doing interviews with both parents and adolescents was helpful in this study. But I am wondering if there were any downsides to doing dyad interviews compared to interviewing them separately.

Author Response

1: This manuscript examined family facilitators, barriers, and strategies of healthy eating among Chinese adolescents using qualitative methods. It is an interesting area of research; however, there are some parts that need clarification.

  1. How many potential participants were approached and how many accepted/ rejected? How was the recruitment process? Please give more details on participants recruitment. Were they compensated?

Response 1: Thank you for the questions. We first contacted 135 eligible families by phone, among whom 30 families did not answer the phone, 57 families refused to participate due to the lack of time or interest and 26 families did not get back after the first contact. We recruited three additional families on-site at a TFES event. Totally 25 families participated in the study. Each family received supermarket coupons of HK$100 to compensate for their time after the interview.  We have added the information in the Subjects session (Lines 92-97).  

  1. The authors mentioned that some participants were part of a larger study. What would be the implication in interpreting the findings besides the volunteer’s bias?

Response 2: Thank you for the question. Family dyads were sampled from families who had enrolled to a comparative cohort study to evaluate an empowerment programme, the Trekkers Family Enhancement Scheme (TFES), targeting low-income families. Health and social empowerment programs such as classes of physical activities and nutrition workshops were provided for voluntary participation by TFES families. (Line 79-83). The potential bias in KAP of healthy eating between our subjects from TFES and other families is acknowledged as a limitation (Line 531-533). However, we could not find any influence from previous participation in nutrition workshops on the interview findings probably because the group was composed of both healthy and unhealthy adolescents.

  1. The data collection period is during the Pandemic from 2020-2022. Is there any chance that this may affect the study's validity?

Response 3: Thank you for the question.  We agree that the eating environment and practices could have been affected by social distancing during the pandemic, but we believe the bias on the results of a qualitative study on long-term family factors should be minimum. We have added this point as a limitation (Lines 535-538).

  1. From personal experience in interviewing adolescents, it was sometimes difficult to let the son/daughter talk openly and honestly in front of their parents. The authors said that doing interviews with both parents and adolescents was helpful in this study. But I am wondering if there were any downsides to doing dyad interviews compared to interviewing them separately.

Response 4: Thank you for the insightful concern. We anticipated that the adolescents might be less willing to talk or be honest in the presence of their parents and strategically invited them to speak before their parents and to start with easy questions on eating practices. We were pleasantly surprised that the adolescents openly described their socially “undesirable” attitudes and practices, such as preference for unhealthy food and selecting fast food and unhealthy snacks on their own, during the interviews (Line 511-516). They were also willing to share how the family motivated and impeded them from eating healthily, with over half of the family facilitators and barriers were agreed between the parents and adolescents (added in Lines 526-528).  It could be explained by the dyads’ mutual perception of the adolescents’ autonomy in food choices, and eating habit is not a sensitive topic.
